# Combination of Self-Healing Butyl Rubber and Natural Rubber Composites for Improving the Stability

**DOI:** 10.3390/polym13030443

**Published:** 2021-01-30

**Authors:** Kunakorn Chumnum, Ekwipoo Kalkornsurapranee, Jobish Johns, Karnda Sengloyluan, Yeampon Nakaramontri

**Affiliations:** 1Sustainable Polymer & Innovative Composite Materials Research Group, Department of Chemistry, Faculty of Science, King Mongkut’s University of Technology Thonburi, Bangkok 10140, Thailand; kunakorn.peach@gmail.com; 2Department of Materials Science and Technology, Faculty of Science, Prince of Songkla University, Songkhla 10400, Thailand; ekwipoo@gmail.com; 3Department of Physics, Rajarajeswari College of Engineering, Bangalore 560074, India; jobish_johns@rediffmail.com; 4Sino-Thai International Rubber College, Prince of Songkla University, Songkhla 10400, Thailand; karnda.seng@gmail.com

**Keywords:** self-healing, natural rubber, carbon nanotubes, electrical recyclability, tire application

## Abstract

The self-healing composites were prepared from the combination of bromobutyl rubber (BIIR) and natural rubber (NR) blends filled with carbon nanotubes (CNT) and carbon black (CB). To reach the optimized self-healing propagation, the BIIR was modified with ionic liquid (IL) and butylimidazole (IM), and blended with NR using the ratios of 70:30 and 80:20 BIIR:NR. Physical and chemical modifications were confirmed from the mixing torque and attenuated total reflection-fourier transform infrared spectroscopy (ATR-FTIR). It was found that the BIIR/NR-CNT_CB_ with IL and IM effectively improved the cure properties with enhanced tensile properties relative to pure BIIR/NR blends. For the healed composites, BIIR/NR-CNT_CB_-IM exhibited superior mechanical and electrical properties due to the existing ionic linkages in rubber matrix. For the abrasion resistances, puncture stress and electrical recyclability were examined to know the possibility of inner liner applications and Taber abrasion with dynamic mechanical properties were elucidated for tire tread applications. Based on the obtained *T_g_* and *Tan δ* values, the composites are proposed for tire applications in the future with a simplified preparation procedure.

## 1. Introduction

Investigations on elastomers for tire application have been attractive in the field of materials research over the past several decades. Particularly, in the case of electric vehicle (EV) technology, the tire with low thickness, high durability and electrically conducting are essential. Nowadays, many types of natural rubber (NR) [1] and synthetic rubber (SR) have been investigated such as styrene-butadiene rubber (SBR) [2], butadiene rubber (BR) [3] and their blends including NR/SBR, SBR/BR and NR/BR/SBR [4,5,6]. This is attributed to the superior elasticity and flexibility of NR and BR [7,8] along with the SBR which has well rolling properties and wet fraction due to π–π interaction of existing benzene ring in the molecular structure [9]. In order to balance the magic triangle of tire application, different types of reinforcing fillers have been used such as clay [10,11,12], silica [13], carbon black (CB) [14,15], nano-calcium carbonate (CaCO_3_) [16,17], graphene (GP) [18,19] and carbon nanotubes (CNT) [20,21,22]. Particularly, the combination of CB/CNT with rubber exhibits superior mechanical, dynamical and electrical properties due to the three-dimensional networks of both fillers in the rubber matrix [23]. 

CNT is an excellent electrically conducting filler used in various rubber products due to its sp^2^ hybridized-carbon structure and high aspect ratio [24,25,26]. Electrons flow easily along the CNT rods and it provides higher conductivity to the composites. However, conductivity of the composites showed 5–6 orders of magnitudes lower than the pure CNT [27,28] due to the formation of bound rubber on the filler surfaces, preventing electron tunneling among end-to-end of CNT particles [29,30]. Hence, the addition of secondary carbon-based fillers, i.e., CB minimizes the thickness of bound rubber layer around the surface of CNT and allows the electron charges for tunneling across the filler networks [31]. With the synergistic effect of CNT and CB, the optimal conductivity of composites has been significantly increased relative to the one with CNT or CB [32]. This phenomenon has observed also in the other polymer matrices, i.e., poly(lactic acid) (PLA) [20], and poly(vinyl chloride) (PVC) [16]. With the addition of CNT, monitoring of dispersion and distribution of CNT inside polymer composites needs to be clarified based on observing modulus, electrical conductivity, rheological properties and morphologies. Zare et al., attempted to explain the formation of CNT network in different matrices using new developing model. The model indicated well the phase separation of the polymer blends filled with CNT and also the formation behavior of the CNT networks. Here, the interphase thickness and the number of filler-filler contacting can be estimated. It was found that the obtained results are significantly correlated to the present of Young’s and storage moduli of the composites which increase with increasing degree of CNT network formation [20,28]. In addition, Ma et al., found that the agglomeration of CNT inside the matrix causes strong lowering of the observed modulus of composites. This was fitted with the micromechanical model which indicated decrease of bound rubber absorption with increasing filler surface area [16]. This means that the key factor for driving the properties of rubber composites is the filler dispersion which significantly relates to the representation of storage and loss moduli, conductivity and mechanical properties.

Thus, rubber composites filled with carbon-based particles were found to be an attractive choice towards the development of new materials suitable for tire applications particularly inner line and tire tread. Recently, self-healing rubber has proposed with movement of positive and negative charges across the cutting interphases. Also, it was found that combination of conductive filler in the composite significantly improves the healing behavior of the composites [33,34]. However, study on self-healing composites for expanding the possibility in tire application has previously not been reported elsewhere and it is a promising and challenging research by improving the electrical conductivity and durability.

Therefore, a self-healing BIIR modified with butylimidazole (IM) was reported in the present work. Due to the increase in durability, 1-butyl-3-methylimidazolium bromide has been selected as an ionic liquid (IL) and the blends of BIIR and NR filled with CNT and CB hybrid fillers were prepared using the ratios of 70:30 and 80:20 phr (BIIR:NR). The work aims to develop healing composites with superior cure time, mechanical and dynamical properties and electrical conductivity. In addition, the puncture stress with electrical recyclability of the composites was examined to determine the healing efficiency. The Taber abrasion resistance and morphological properties were also reported. The goal is to promote the usage of such composites for inner line and tire tread applications.

## 2. Materials and Methods

### 2.1. Materials

Bromobutyl rubber (BIIR) was manufactured by ExxonMobil Co., Ltd. (Irving, TX, USA). Natural rubber (NR) received as Standard Thai Rubber (STR 5L) was manufactured in a local factory operated by Na Born Farmer Cooperation (Nakorn Si Thammarat, Thailand). The multi-wall carbon nanotubes (CNT) were used with 9.5 nm in diameter, ca. 1.5 µm length and 90% purity. CNT of NC7000 type was manufactured by Nanocyl S.A. (Sambreville, Belgium). Carbon black (CB), Vulcan XC72, with 30 nm diameter was manufactured by Cabot Corporation (Pampa, TX, USA). 1-butylimidazole (IM) and 1-butyl-3-methylimidazolium bromide (IL) were purchased from Merck KGaA. (Darmstadt, Germany). In addition, stearic acid was procured from Imperial Chemical Co. Ltd., (Pathumthani, Thailand). Zinc oxide (ZnO) and sulfur were purchased from Global Chemical Co., Ltd. (Samutprakarn, Thailand) and Ajax Chemical Co. Ltd. (Samutprakarn, Thailand) respectively. 2,2′-Dithiobis-(benzothiazole) (MBTS) was supplied by Flexsys Inc., Termoli, Italy. 

### 2.2. Preparation of Self-Healing BIIR/NR Composites

Preparation of the composites was carried out with three different steps following the chemical formulation as shown in Table 1. Initially, the modification of BIIR was done by using an internal mixer (Brabender VR GmbH & Co. KG, Duisburg, Germany) at 40 °C and 60 rpm rotor speed. Here, BIIR was masticated for 3 min before adding IL or IM at 5 and 10 phr. The mixing was continued up to 15 min for promoting excess crosslinking. In case of pure BIIR, the rubber was solely masticated for 15 min for comparison purpose. Pure BIIR and modified-BIIR were kept in a desiccator for at least 2 h and labeled as COMPOUND I in Table 1 and Figure 1. Another part was the compounding of NR composites with CNT-CB hybrid fillers using an internal mixer and open two-roll mill (Charoen Tut Co., Ltd., Samutprakarn, Thailand) under controlled conditions of 80 °C and 60 rpm. Before compounding, the filler (CNT:CB = 5:7.5 phr [23]) was mixed with ethanol in the ratio of 1:10 *w*/*v* using ultrasonication at 80 W for 10 min. The mixing operation was then started by masticating NR in an internal mixer for 1 min before adding the filler dispersed in ethanol, and the mixing continued for another 6 min. Here, the ethanol was directly evaporated during mixing. The activators (i.e., stearic acid and ZnO) and the curatives (i.e., MBTS and sulfur) were consecutively added into the rubber compound and the mixing was continued until a total mixing time of 12 min. The pure NR was prepared by the same procedure and mixing time for comparison purpose. The compounds were then stored in desiccator for at least 2 h and labeled as COMPOUND II in Table 1 and Figure 1. Later, COMPOUNDS I and II were masticated individually on two-roll mill for controlling the initial Mooney viscosity of 55–60 ML1 + 4(100 °C). Eventually, COMPOUNDS I and II were mixed using an internal mixer at 40 °C and 60 rpm for 10 min at the ratios of 70:30 and 80:20 phr. The resulting compounds were passed through the 1 mm nip of the two-roll mill for several times to achieve optimal dispersion of the filler in the rubber matrix and kept in a desiccator for at least 24 h. Finally, the rubber composite sheets with dimensions 150 mm × 160 mm × 2 mm were prepared by compression molding at 160 °C using the cure times based on rheometer tests.

### 2.3. Characterization

#### 2.3.1. Attenuated Total Reflection-Fourier Transform Infrared Spectroscopy (ATR-FTIR)

The ATR-FTIR spectra were recorded by using ThermoNicolet Avatar 360 FTIR (Thermo Electron Corporation, Madison, WI, USA), with 4 cm^−1^ resolution and 64 scans per sample. The instrument was equipped with a germanium ATR crystal probe. The functional groups present in modified and un-modified BIIR with and without blending with NR composites were investigated in order to clarify the chemical and physical interactions in the composites. 

#### 2.3.2. Cure Characteristics

Cure characteristics of pure rubbers and their composites with CNT-CB hybrid filler were determined using a moving die rheometer (MDR, Monsanto Co., Ltd., Findlay, OH, USA). The measurements were performed at a fixed oscillating frequency of 1.66 Hz with 1 arc degree amplitude at 160 °C. Here, the scorch time (*T_s_*_1_) refers to the time that the torque increase for 1 dNm based on the minimal torque. The *T*_90_ is the cure time of the sample and *M_H_–M_L_* is the difference between maximal and minimal torques as a function of time.

#### 2.3.3. Tensile Properties

The tensile tests were performed at 23 ± 2 °C using a Zwick Z 1545 tensile testing machine (Zwick GmbH & Co. KG, Ulm, Germany). The five dumbbell-shaped specimens were used according to ISO 527 (type 5A) at a cross-head speed of 200 mm/min. For the self-healing tests, the dumbbell-shaped samples were quickly cut with a sharp razor blade (Energizer^®^ Holdings, Inc., St. Louis, MI, USA) and kept in the same dumbbell-shaped mold. Then, the entire setup with sample and mold was heated in a hot air oven at 120 °C for 30 min without applying any pressure. The healed samples were removed, kept in desiccator for 24 h and performed the tensile test with the same cross-head speed of 200 mm/min. In addition, the puncture test was carried out with a sample size of 100 mm × 20 mm × 2 mm and the amount of force for puncturing the samples was detected through the needle tip.

In order to examine filler-rubber interaction the bound rubber thickness was investigated. Rectangular test pieces (dimension of 4 mm × 2 mm × 25 mm) of the samples before performing the compression molding were immersed in 100 mL toluene for 7 days at room temperature. The rubber–filler gel was then carefully separated from the mixture by filtration before drying at 70 °C for 3 h (until constant weight). The average thickness of rubber layer bound to the filler surfaces (***δ****′*) was calculated using Equation (1) [23]:(1)$$\delta^{\prime }=\;\frac{m_{2}-\;m_{1}\left(C_{f}\right)}{\rho_{R}m_{1}\left(C_{f}\right)S_{f}}$$
where ***m***_1_ is the mass of the rubber compound before extracting, ***m***_2_ is the mass of the rubber–filler gel consisting of the non-dissolving bound rubber part and the filler. Parameters ***C_f_*** and ***S_f_*** are the mass concentration and the specific surface area of filler in the composites, respectively, whereas ***ρ_R_*** refers to the density of BIIR/NR matrix (i.e., ***S_f_*_CNT_**~540 m^2^/g and ***S_f_*_CB_**~254 m^2^/g).

#### 2.3.4. Payne Effect

Payne effect is usually attributed to the filler flocculation in the polymer matrix. High Payne effect means strong filler agglomeration, indicating poor reinforcing efficiency in the matrix. The Payne effect can be estimated from the storage moduli at maximal and minimal strain amplitudes [35]. In the present work, the Payne effect was determined by using a rubber process analyzer (RPA) (Alpha Technologies, Akron, OH, USA), which measures the storage shear moduli (*ε’*) of filled BIIR/NR compounds cured under shear deformation. Payne effect value was calculated from the difference of *ε’* at maximal and minimal strain sweep of 0.56 and 100% at 1 Hz oscillating frequency and 100 °C.

#### 2.3.5. Morphologies

The surface morphologies of pure rubber and their composites were characterized by optical microscopy (Carl Zeiss Microscopy GmbH, Oberkochen, Germany). The rubber samples were quickly cut with a razor blade in order to create a smooth surface before capturing the microscopic photographs. Furthermore, the samples were characterized by transmission electron microscopy (TEM), (Jeol JEM 2010, Jeol Ltd., Tokyo, Japan) using 200 kV acceleration voltage. Ultrathin slices (about 80–90 nm) were prepared using diamond knife in the RMC-MT-X ultramicrotome (Boeckeler Instruments, Inc., Arizona, AZ, USA), under cryogenic conditions at −100 °C. The thin samples were then mounted on 200 mesh copper grids before imaging by TEM. 

#### 2.3.6. Electrical Conductivity

Electrical properties of the samples were measured at room temperature using an LCR meter (Hioki IM 3533, Hioki E.E. Corporation, Nagano, Japan) by means of resistance (***R_P_***) at a frequency of 50 Hz. The sample of 100 mm × 100 mm × 2 mm was first placed between two parallel plates of the dielectric test fixture (16451B dielectric test fixture, Test equipment Solutions Ltd., Berkshire, UK) with 5 mm electrode diameter. The electrical conductivity (*σ*) was calculated by using the following equation [36]:(2)$$\sigma =\frac{1}{\rho }=\frac{d}{\left(R_{p}\right)A}$$
where ***d*** and ***A*** refer to the sample thickness and the area of the electrode, respectively. The factor ***ρ*** is the volume resistivity, the reciprocal of conductivity.

For the electrical recyclability, the same samples were cut into 10 mm × 10 mm × 2 mm before re-compressing for three different cycles. For each cycle the conductivity was measured in order to clarify self-healing efficiency of the composites.

#### 2.3.7. Abrasion Resistance

Table abrasion by means of wear index was used to elucidate the abrasion resistance of the self-healing composites. The sample with diameter of 110 mm was placed into the Table abrasion tester (Model GT-7012-T, GoTech, Taichung, Taiwan) with 70 rpm rolling speed for 1000 rounds (three samples/formula). Here, the wear index was calculated by using the equation:Wear index (%) = (*W_b_* − *W_a_*) × 100/*W_b_*(3)
where *W_a_* and *W_b_* refer to the weight of the sample before and after performing the Taber abrasion tests.

#### 2.3.8. Dynamic Mechanical Analysis

Dynamic mechanical behavior of the samples was determined using DMA 1 (Mettler-Toledo GmbH., Zurich, Switzerland). The experiment was carried out in tension mode with a frequency of 10 Hz at a strain of 0.2%. The dynamic properties in terms of storage modulus, loss modulus, and loss factor (*Tan δ*) were scanned with a heating rate of 2 °C/min over the temperature range from −90 to 80 °C in order to characterize the elasticity of the self-healing composites.

## 3. Results and Discussion

### 3.1. Interactions of IL and IM with Rubber Molecules

Physical and chemical interactions among IL and IM molecules and the BIIR/NR molecular chains are clarified from the mixing torque and ATR-FTIR of the compounds and vulcanizates. Figure 2A shows the mixing torque of BIIR modified with 5 and 10 phr of IL and IM relative to the pure BIIR. The mixing torque of the compounds is increased particularly after the addition of IM, whereas a drastic reduction is observed in case of other samples. It means that the chemical crosslinking among BIIR macromolecules is occurred under the combined influence of temperature and shear force in terms of nirogen-alkylation reaction [33] as shown in the proposed reaction (Figure 2). During mixing operation, the temperature arises from chain scission of BIIR macromolecules activates the nucleophilic aliphatic substitution between alkyl halide in BIIR and amine in IM. This forms a quaternary ammonium salt in the Menshutkin reaction with N-cation and Br-anion [37]. The reaction can be affirmed with the ATR-FTIR peaks, as seen in sub-Figure 2A. Specific peaks at 670 and 1164 cm^−1^ which are assigned to the H-C-C and H-C-N bending vibrations [38] from the existence of IM and IL imidazole structures and the -C-Br stretching vibrations from BIIR chains, respectively [38,39]. The peak at 1164 cm^−1^ is found to be disappeared after the addition of IM, whereas it is still distinguished in IL case. This means that the addition of IM produces chemical reaction with BIIR molecular chains at the position of Br branching, as shown in Figure 2. Figure 2A shows that the addition of 10 phr IM showed significantly lower mixing torques than the one with 5 phr. With excess of IM loading, the excess IM might act as a plasticizer in the compound and induce the movement of rubber chain during mastication. On the other hand, the IL might physically cover the BIIR macromolecules since the mixing torque is slightly decreased with time. Therefore, it summarizes that the modification of BIIR with IM generates positive and negative charges on BIIR macromolecules, whereas the addition of IL cannot originate the chemical reaction with the main chain of BIIR.

After the combination of pure BIIR and modified-BIIR with NR composites filled with CNT and CB hybrid filler (BIIR/NR-CNT_CB_), the ATR-FTIR is shown in Figure 2B. As expected, there is no significant difference in the absorption peaks between the blend ratios 70:30 and 80:20 of BIIR:NR. Also, the specific peaks of NR are appeared at 832, 1375, 1449, and 1538 cm^−1^ which attributed to =C-H out of plan bending, asymmetric and symmetric C-H stretching, C=O stretching of zinc stearate and C=C stretching vibrations of isoprene unit, respectively [29,30]. However, there is no shift of the peak at 1164 cm^−1^, while the peak at 670 cm^-1^ is completely disappeared particularly for the one with 5 phr IM. This clarifies that there is no chemical interaction among BIIR/NR and IL. Thus, a slight decrease in the peak intensity at 1164 cm^−1^ is explained by the physical absorption of IL on CNT and CB surfaces [39]. According to the π–π interaction of CNT-CB and IL, IL is well absorbed by the filler surfaces and forms electron bridges among CNT-CB-CNT networks [40].

### 3.2. Cure Characteristics

Figure 3 shows the cure characteristic curves of modified and unmodified rubbers along with their composites prepared by reinforcing CNT-CB hybrid fillers. Similarly, Table 2 summarizes the cure characteristics of the compounds and vulcanizates. It is also noted that the same behavior is observed for the blend ratios of 70:30 and 80:20. Here, the pure BIIR clearly exhibited non-crosslinking by drastic decrease in torques with respect to time. For the pure NR, vulcanization reaction with sulfur curing system is reached, but with strong reversal behavior from thermo-oxidative degradation relating to the existing C=C double bonds [41]. In contrast, marching curves (i.e., increasing torque with time beyond 90% cure propagation) are seen in the BIIR/NR composites without IL and IM. This is due to the different curing rate of NR and BIIR with sulfur atoms since the steric hindrance of isobutylene in BIIR molecular chains. This shields the reaction of sulfur atoms to allylic carbon position in the isoprene units [42]. Thus, the vulcanization reaction initiated fast and with a slow curing rate. In Figure 3, faster curing of the composites with CNT-CB is observed after the modification of BIIR macromolecules with IL and IM. Also, strong reversal and marching cure curves are minimized. Considering *T_s1_* and *T_90_* of the composites, both the specific times were strongly decreased when IL and IM were added relative to the pure rubbers and the ones without modifiers. This is due to (I) improved CNT-CB dispersion and distribution throughout BIIR/NR matrix, (II) superior thermal conductivity of CNT, CB, IL, and IM [43] and (III) catalytic effect of IL for rubber crosslinking origination [44]. Due to the presence of liquid phases in the compounding, breakages of filler agglomeration during mixing operation are initiated. As expected, from the extremely high thermal conductivity of CNT and CB, heat flows easily throughout the matrix and the curing process completes soon. With the absorption of IL and IM on the filler surfaces the heat flow might be faster, therefore the values of *T_s1_* and *T_90_* are found to be lowered [45]. 

Considering the *M_H_*–*M_L_*, which is known as the crosslink density originated in the compounds, it is seen that *M_H_*–*M_L_* increases upon adding CNT-CB and IL, while the crosslink density decreases for the one with IM, particularly in case of BIIR:NR with 80:20 ratio. It shows the prevention of the crosslinking propagation in BIIR/NR with sulfur and IM as the two different crosslinkers. Here, most of the chemical crosslinks are originated by IM. After the incorporation with NR compounds, although the crosslinks can be reversed, the sulfur do not react completely with all the isoprene units. Therefore, the estimated crosslink density of the composites is found to be low, since there is no significant difference between *M_H_* and *M_L_* which prevents sulfur crosslinking from BIIR-IM linkages. Therefore, it can be concluded that the combination of modified-BIIR with IM and IL together with CNT-CB particles showed a strong improvement in the productivity of the material due to the significantly decreased *T_s_*_1_ and *T*_90_ values. Also, the observed vulcanizates do not show reversion and marching behaviors even a slight decrease of *M_H_-M_L_* is found.

### 3.3. Relations of Mechanical, Dynamic Mechanical, and Morphologies 

#### 3.3.1. Before Self-Healing Propagation

Figure 4 shows the comparison of stress–strain curves of pure rubber with and without IL and IM modifiers along with their composites. Also, the results of 100% modulus, tensile strength and elongation at break are summarized in Table 3. It is seen that BIIR/NR-CNT_CB_ composites exhibited higher modulus at 100% elongation and tensile strength than pure rubbers, while the elongation at break is reduced. This is attributed to the reinforcement efficiency of the carbon-based fillers existing in the BIIR/NR matrices. It is also observed that the ratio of 70:30 shows higher tensile properties than the one with 80:20 due to the higher loading level of CNT-CB hybrid filler into the matrix. Figure 4 and Table 3 show that the addition of IL and IM significantly changed the properties of composites. For the composites with IL, improved modulus and lowered tensile strength and elongation at break are found. It must be noted that the improved modulus is due to the plasticizing effects of IL increases the rubber chain entanglement. However, existing IL (i.e., liquid phases) in the matrix makes more defects during extension and causes the material easily breakable, lowers the tensile strength and elongation [45,46]. In case of IM, entirely different results have been observed when compared to IL. It is seen that the incorporation of IM reduces the modulus and the tensile strength than IL-based composites of both 70:30 and 80:20 ratios of BIIR:NR. It clearly correlates with the poor crosslink density as shown in Figure 3. Preventing sulfur attraction due to N-alkylation of BIIR-IM-BIIR linkages prohibited the complete process of crosslinking for NR and BIIR macromolecules during vulcanization. This leads the molecular chains to break easily and limits the degree of filler dispersion. However, according to the existing ionic crosslinks in rubber chains, interaction among the breakage of chains might be the reasons of superior elongation at break. In other words, after the cleavage of rubber chain entanglement, molecular weight of the composites is strongly decreased and moved following the extension direction. During extension, ionic interaction among the chains causes healing and resisting the final breakages of the samples, indicating high elongation efficiency.

Effects of the filler dispersion throughout the BIIR/NR matrices can be elucidated by relating storage shear modulus versus strain amplitude as displayed in Figure 5. As a result, the Payne effect is calculated by referring the filler flocculation and showed in Figure 6. Considering in Figure 5, it is clearly seen that the storage modulus of the BIIR, NR and BIIR/NR increases with the addition of CNT-CB hybrid filler. This is due to the reinforcement efficiency of the filler to rubber matrix. In addition, it is seen that the highest storage modulus had found from the composites with IM, while the one with IL showed poor properties. This affirms the efficiency of the chemical crosslinking among BIIR macromolecules after mixing with NR which is also the rationale of slight higher storage modulus of the composites using BIIR:NR ratio of 80:20 than 70:30. According to the observed Payne effect in Figure 6, it is noted that the Payne effect of pure rubbers is due to the agglomeration of CNT-CB particles in the matrix. As indicated in the previous work, the CB had the agglomerate size of approximately 200 nm in the rubber matrices, whereas only 10 nm is found for the CNT diameter [23]. In addition, as discussed in the stress–strain curves, the low Payne effect is well related to the dispersion of filler particles presented for the composites with IL at the ratios of 70:30 and 80:20. It clarifies the improved filler distribution in BIIR/NR due to the presence of liquid phase during the process of mixing. Self-crosslinking occurs in the BIIR/NR matrix increases agglomeration of CNT-CB and exhibited high Payne effect related to storage shear modulus under strain amplitudes. Furthermore, filler agglomeration in BIIR/NR matrix can be seen in the morphologies as shown in Figure 7. It is seen that the BIIR/NR-CNT_CB_ and BIIR/NR-CNT_CB_-IM exhibited high surface roughness due to strong filler clusters in both 70:30 and 80:20 ratios. The strongest CNT-CB agglomeration occurs in case of 80:20. This can be related to high steric hindrance of BIIR which reduces the filler dispersion. On the other hand, the BIIR/NR-CNT_CB_ with IL showed a smooth surface with homogenous dispersion of the CNT-CB hybrid filler. Considering the indication of storage modulus of the composites, relating results with the observed tensile strength and elongation at break are shown. This is contributed by the rubber–filler interaction by means of interfacial adhesion of BIIR/NR macromolecules and CNT-CB surfaces [11]. Figure 8 shows the bound rubber thickness of the composites which indicated opportunity of rubber and filler to interact together. With the addition of CB inside CNT-based matrix, the bound rubber layer decreases significantly and this increases effectively the rubber–filler interaction [14,17]. However, the thickness had relatively increased with the combination of conductive NR to BIIR at 70:30 and 80:20 phr (2.82 and 3.47 nm). This causes high rubber–filler interaction in 70:30 of BIIR/NR-CNT_CB_-IM_5_, as also seen in the sub-Figure 8 relating the TEM images. This is the rationale of enhanced modulus, tensile strength and storage modulus of the composites after the addition of CB using 70:30. The result has correlated well with the modeling of Zare et al. which studied the formation of CNT network inside several polymer matrices. Formation as the three dimensional CNT pathway induced strongly the mechanical properties of the composites following the reinforcement mechanism [47]. This is in good agreement with the observed superior modulus (Figure 4) and low Payne effect (Figure 6) of the composites.

#### 3.3.2. After Self-Healing Propagation

With the aim of inducing abrasion resistance of the composites, possibilities of self-healing of the composites are investigated and examined. Here, all the dumbbell samples were cut quickly, added to the mold and heat at 120 °C for 30 min without applying pressure. Figure 9 presents the stress–strain curves of the healed-composites, whereas the values of modulus, tensile strength and elongation at break are depicted in Table 4. It is found that all the composites with BIIR showed potentially self-healing, especially the ones with IL and IM modifiers. Slight lowering of 100% modulus is found for BIIR/NR, BIIR/NR-CNT_CB_, and BIIR/NR-CNT_CB_-IM with 70:30 and 80:20 ratios. The occurrence of reversible crosslinking in terms of chain entanglement and ionic crosslinks can be explained with the help of the proposed model as shown in Figure 10. In case of IL-based composites, the physical covering of IL molecules around BIIR/NR macromolecules and CNT/CB surfaces can convey rubber molecular chain to produce re-entanglement at the cutting interfacial area (Figure 10A). As expected, due to the existence of liquid phase inside the rubber matrix, the lowest tensile properties are observed. On the other hand, in case of IM-based composites, chemical grafting of IM on BIIR molecular chains causes strong movement of the molecules to re-bond through ionic linkages (Figure 10B). This is a rotational and superior property of BIIR/NR-CNT_CB_-IM after healing propagation. Finally, the composites showed the highest modulus due to the attraction of positive and negative charges present on its molecular main chain [34,44]. 

Topologies of the healed composites are shown in Figure 11. At the joint of BIIR/NR-CNT_CB_, a black area can be observed which refers to the depth in the composites. This means that the composites have not been completed the healing process at 120 °C for 30 min. On the other hand, in the case of BIIR/NR-CNT_CB_-IL and BIIR/NR-CNT_CB_-IM composites, a homogenous healing of the composites can be achieved due to the excellent healing process as proposed in the model (Figure 10). 

### 3.4. Durability of the Composites

The durability of the present BIIR/NR composites filled with CNT-CB before and after modification with IL and IM has been examined for the purpose of inner liner and tire treads to be used in tire application. Consequently, the puncture stress with electrical recyclability and abrasion resistance along with dynamic mechanical properties of the composites are investigated and discussed.

#### 3.4.1. Relation between Puncture Stress and Electrical Conduction Recyclability

The puncture force is estimated as the force that requires for puncturing the composites samples. This is used for stimulating the attraction of the object to the inner liner in tires. Figure 12 shows the puncture stress of pure rubbers and their composites. Pure NR exhibited higher stress compared to BIIR, and the value increases on adding the CNT-CB hybrid filler. This is due to the reinforcement efficiency of both the carbon-based fillers in the BIIR and NR matrices. However, in case of both the BIIR/NR ratios, the puncture stress strongly reduced upon the addition of IL. Although, the IL improves the CNT-CB dispersion in BIIR/NR, existing defect points of the liquid phases reduce the breaking resistance of the composites effectively [46,47]. Hence, the crack propagation occurs rapidly and the composites can break easily. On the other hand, using of IM modifier strongly resists the breakage of matrix by the formation of chemical linkages from sulfur and ionic crosslinking [34]. The BIIR:NR of 70:30 shows the highest puncture stress due to the superior chain entanglement of the NR and the reinforcing effects of CNT-CB. It summarizes that the combination of modified-BIIR with IM and NR-CNT_CB_ composites increases the puncture resistance from the attack of any foreign objects. 

However, repeatability of the puncture test after self-healing is not straightforward and thus the electrical conductivity of the composites has been investigated. The compressed sample sheets are cut in to small pieces and performed a re-compression molding up to a maximum of three cycles before measuring the conductivity individually. Figure 12 shows the results of electrical conductivity of the composites relative to the pure ones. It is clearly seen that the addition of CNT_CB_ enhances the conductivity of the composites and reached the maximum conductivity on the incorporation of CNT_CB_ along with IL. As expected, it is a result of improved filler dispersion inside the rubber matrix. In the presence of IM, a well dispersed form might prohibit the three dimensional networking of rubber macromolecules through sulfur and IM crosslinking [48,49]. The same trend can be observed for BIIR:NR with the ratios of 70:30 and 80:20 (lower than 1 order of magnitude differences). As seen in Figure 8, addition of CB reduced the bound rubber thickness of the composites to lower than 4 nm, and therefore the electron charges can tunnel across the interphase between BIIR/NR chain and CNT-CB surfaces [18,23,49]. In Figure 13, it can be seen that the composites reached the percolation threshold since the conductivity increased significantly with addition of CNT-CB hybrid filler and changed the insulator BIIR/NR to be the semi-conductor. Thus, it is realized here that the using of CNT-CB forms well the filler pathway in the rubber matrix and the electron can easily jump through the filler-rubber–filler throughout the composites [12,45]. Interestingly, the trend has changed significantly on performing the compression recyclability. It is found that the BIIR/NR-CNT_CB_-IM showed only a slight decrease in the conductivity after three cyclic-conductivity tests. This means that the healing propagation of the composites causes not only the re-bonding of the rubber molecular chains, but also develops filler re-networking. It clarifies well from the healing efficiency of the composites with IM modifier which increases the strength in terms of tensile properties (Figure 4 and Figure 9) and puncture stress (Figure 12).

#### 3.4.2. Relation between Abrasion Resistance and Dynamic Mechanical Properties

Figure 14 shows the Taber abrasion resistance of the composites by means of wear index indication. The low wear index indicates the superior abrasion resistance. The same trends of abrasion are observed for the composites of BIIR:NR with the ratios of 70:30 and 80:20. With the same range of Mooney viscosity, it hypothesized that the molecular weight of both pure NR and BIIR do not show significant difference. The pure BIIR shows a lower wear index than NR due to the interlocking of steric branching on their molecular chain. It is also seen that the values are relatively decreased on the addition of CNT-CB fillers. It can be related to the bounded rubber on the filler surfaces that reduces abrasion. This restricts the movement of rubber chains and increases the abrasion resistance of the composites. It is seen in Figure 14 that the lowest wear index is found for the BIIR/NR-CNT_CB_-IM. It clarifies the superior self-healing of the composites that prevents well the abrasion behavior, indicating superior wear index within the range of tire tread used in passenger cars [50]. Surface images of the composites after the Taber abrasion testing are presented in Figure 15. The highest roughness surface is observed in pure NR followed by pure BIIR and BIIR/NR blends, respectively. Smoother surfaces are observed in the composites with CNT-CB hybrid filler, while the most flat surface is observed for the composite BIIR/NR-CNT_CB_-IM_5_. It correlates well with the observed Taber abrasion resistance of the composites as shown in Figure 14. However, according to the magic triangle of tire application, it is not only the abrasion resistance of the composites, but the wet fraction and rolling resistance have also to be considered. Figure 16 shows the relation between storage modulus and *Tan δ* of pure rubbers and their composites as a function of temperature. Also, the glass transition temperature (*T_g_*), maximal Tan δ (*Tan δ_max_*) and *Tan δ* at 0 and 60 °C (*Tan δ*_0_ and *Tan δ*_60_) are summarized in Table 5. The *T_g_* of composites is found to be in between the *T_g_* of NR and BIIR. There is no considerable change in the *T_g_* after the addition of CNT_CB_, CNT_CB_-IL, and CNT_CB_-IM. In addition, the value of *Tan δ_max_* indicates the degree of chain mobility and damping properties. The highest *Tan δ_max_* is noticed in pure NR and the value decreases significantly after the addition of fillers and modifiers. The lowest *Tan δ_max_* is observed for the composites with CNT_CB_-IM. This is attributed to the crosslinking of matrix phase in the composites from sulfur and IM, and limited the mobility of rubber molecular chain. The lowest crosslink density of BIIR/NR-CNT_CB_-IM composites is due to the IM already crosslinked within the composites before performing the curing studies. For the IL-based composites, higher *Tan δ_max_* value is observed in case of composites with CNT_CB_ and CNT_CB_-IM. This is due to the plasticizing effect of the existing IL in the composites which increases the chain mobility and decreases the damping properties of the composites [51]. 

In case of *Tan δ*, the value at approximately 0 °C is used to predict the wet traction of a tire, while the one at 60 °C indicates the loss of energy of rubber composite under dynamic deformation which relates to tire’s rolling resistance [52]. At 60 °C, high *Tan δ_60_* refers to low elasticity with low rolling resistance. Thus, the addition of CNT-CB hybrid filler decreases the *Tan δ_60_* owing to the existing bound rubber which increases the elasticity at high temperature of the composites. However, the values have no significant differences among the composites of BIIR/NR-CNT_CB_, BIIR/NR-CNT_CB_-IL and BIIR/NR- CNT_CB_-IM. Interestingly, in case of *Tan δ* at 0 °C, different trends are noticed. For BIIR/NR-CNT_CB_ and BIIR/NR-CNT_CB_-IM composites, increasing *Tan δ_0_* is exhibited due to the formation of brittle nature of composites at low temperature. It increases the wet traction considerably which means by increasing the driving ability on the wet surface. On the other hand, in case of BIIR/NR-CNT_CB_-IL, the lowering of *Tan δ_0_* is observed due to the improved filler-rubber interaction because of the well dispersed CNT_CB_ in BIIR/NR matrices. It confirms the improvement in the durability of the composites based on the combination of self-healing BIIR and NR-CNT_CB_ composites as seen in the developed puncture stress (Figure 12), electrical conduction recyclability (Figure 13) and dynamic mechanical properties (Figure 16).

## 4. Conclusions

The self-healing composites based on modified-BIIR and NR-CNT_CB_ composites are successfully prepared using three steps of mixing through an internal mixer and a two-roll mill. The results showed that, IL and IM exhibited different modification models to BIIR macromolecules. BIIR exhibited a physical cover of IL molecules, while the chemical modification of BIIR chain with IM is also generated. With the ionic interaction among BIIR and IM, the healing propagation shows a fast and strong crosslinking at the interface of cutting area, which is clarified by the superior tensile properties of the healed-composites. The present work exhibited that the addition of IL and IM improved the cure characteristics significantly with short term production. The possibility of using the resulting composites in inner liner application, the self-healing composites showed superior puncture stress. Although the electrical conductivity of BIIR/NR-CNT_CB_-IM exhibited lower when compared to the one with IL. The electrical recyclability is not significantly changed after three cyclic compressing. Regarding the introduction towards the tire tread applications, Taber abrasion of BIIR/NR-CNT_CB_-IM showed the lowest abrasion resistance with superior rolling resistance and wet traction corresponding to the observed *Tan δ_60_* and *Tan δ_0_*. Overall, the healing composites of BIIR/NR with IM at a ratio 70:30 promoted proper properties in terms of curing propagation, mechanical healing, morphologies, electrical conductivity, and recyclability together with suitable abrasion, rolling resistance, and wet traction. It supports, the future of EV tire application which requires simple preparation process, short production procedure, superior healing propagation with semi electrical conductivity, recyclability, high duration properties in terms of abrasion resistance, and varying elasticity at high and low service temperatures.

## Figures and Tables

**Figure 1 polymers-13-00443-f001:**
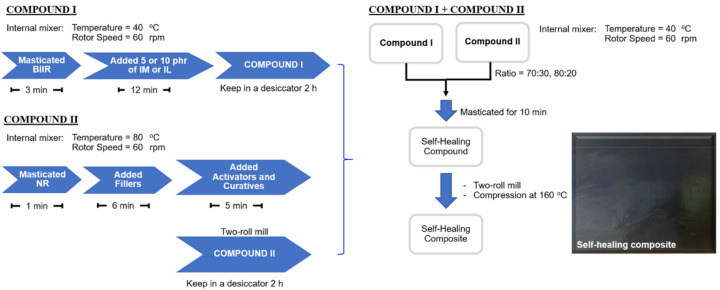
Preparation procedures of the bromobutyl rubber (BIIR)/ natural rubber (NR) composites with ionic liquid (IL) and butylimidazole (IM).

**Figure 2 polymers-13-00443-f002:**
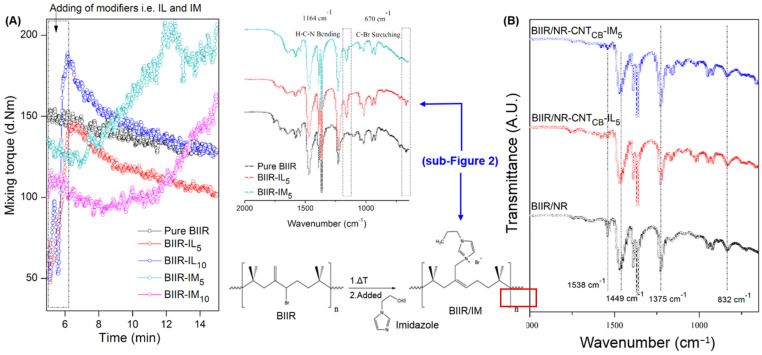
Mixing torque of pure BIIR and modified-BIIR with IL and IM compound (**A**), the ATR-FTIR images and their proposed reaction (sub-Figure 2), and the spectra of the composites with 5 phr of IL and IM (**B**).

**Figure 3 polymers-13-00443-f003:**
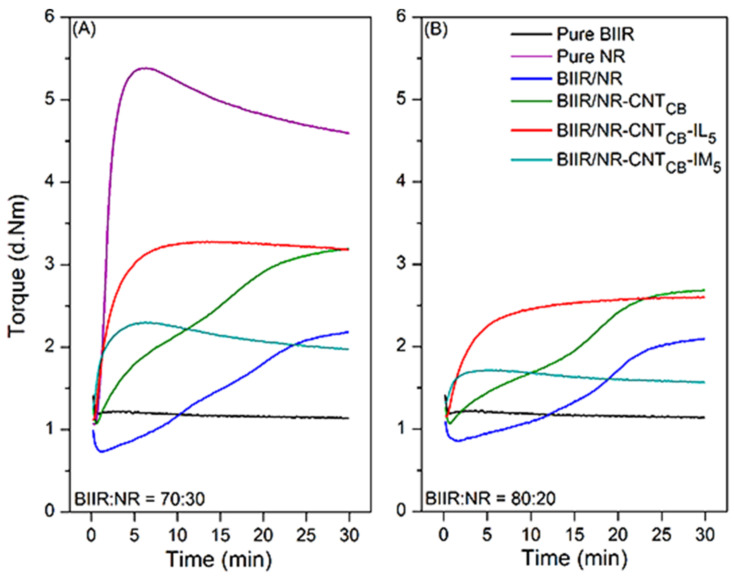
Cure characteristics curves of pure rubbers and their composites with and without IL and IM modifiers, BIIR:NR ratios of 70:30 (**A**) and 80:20 (**B**).

**Figure 4 polymers-13-00443-f004:**
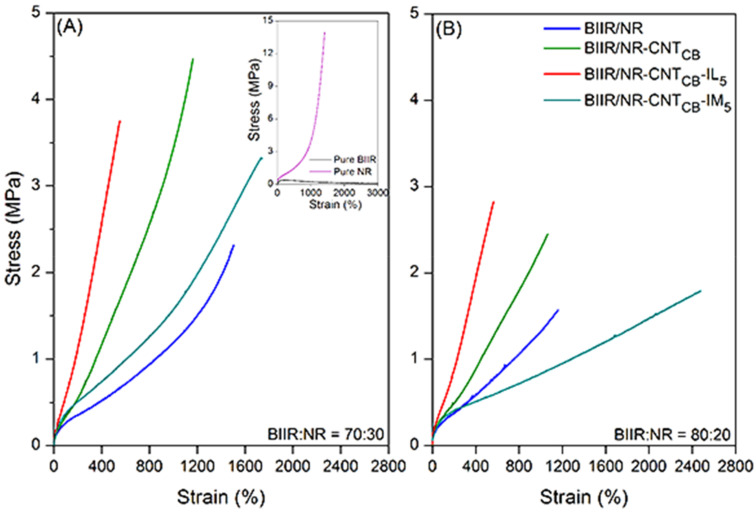
Stress–strain curves of pure rubbers and their composites with and without IL and IM modifiers, BIIR:NR ratios of 70:30 (**A**) and 80:20 (**B**) before healing propagation.

**Figure 5 polymers-13-00443-f005:**
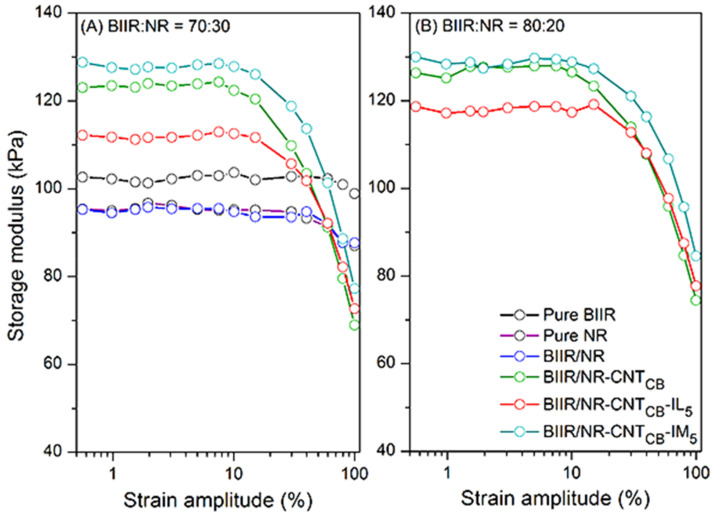
Relation of storage modulus and strain amplitude for pure rubbers and their composites with and without IL and IM modifiers, BIIR:NR ratios of 70:30 (**A**) and 80:20 (**B**).

**Figure 6 polymers-13-00443-f006:**
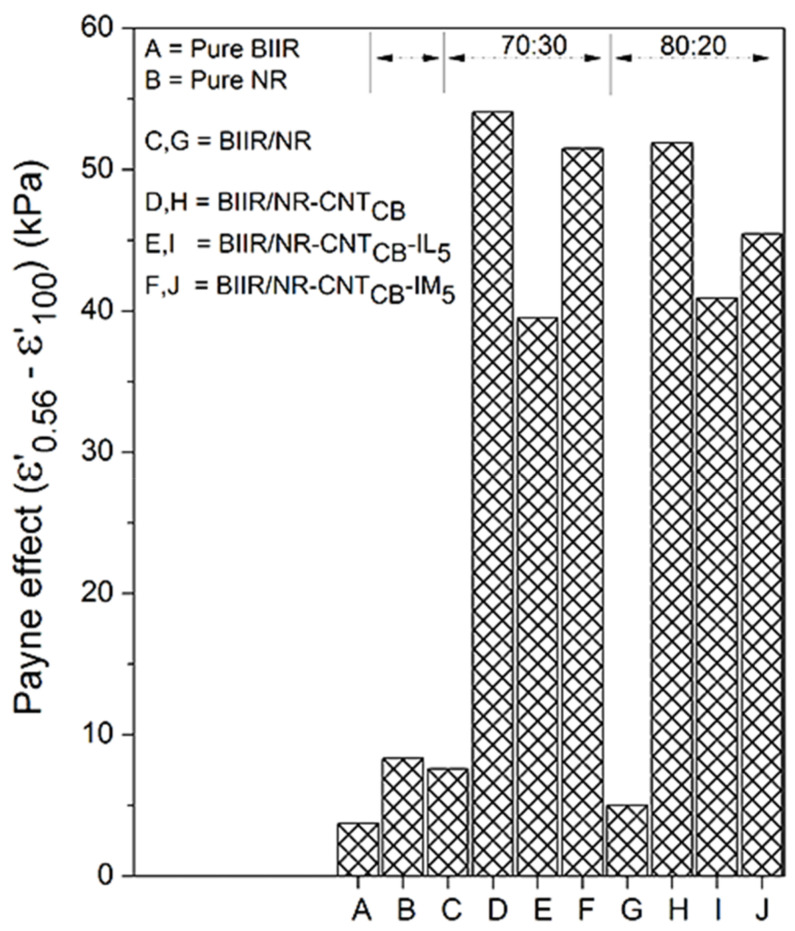
Payne effect of pure rubbers and their composites with and without IL and IM modifiers, BIIR:NR ratios of 70:30 and 80:20.

**Figure 7 polymers-13-00443-f007:**
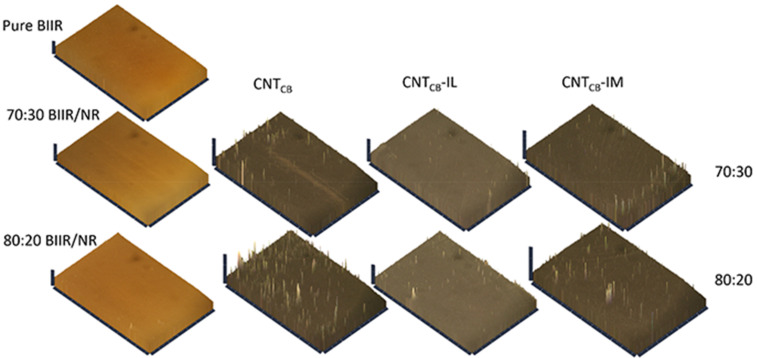
Morphologies of pure rubbers and their composites with and without IL and IM modifiers, BIIR:NR ratios of 70:30 and 80:20 before healing propagation.

**Figure 8 polymers-13-00443-f008:**
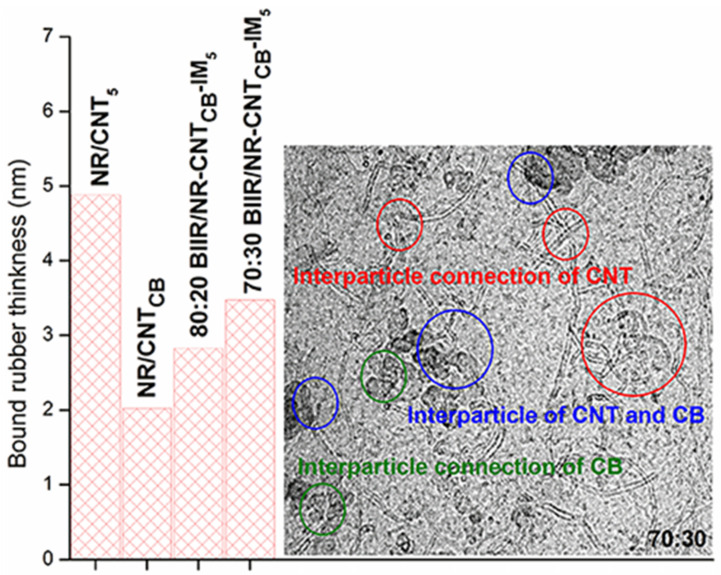
Relation of bound rubber thickness of the BIIR/NR-CNT_CB_ composites filled with IM, BIIR:NR ratios of 70:30 and 80:20.

**Figure 9 polymers-13-00443-f009:**
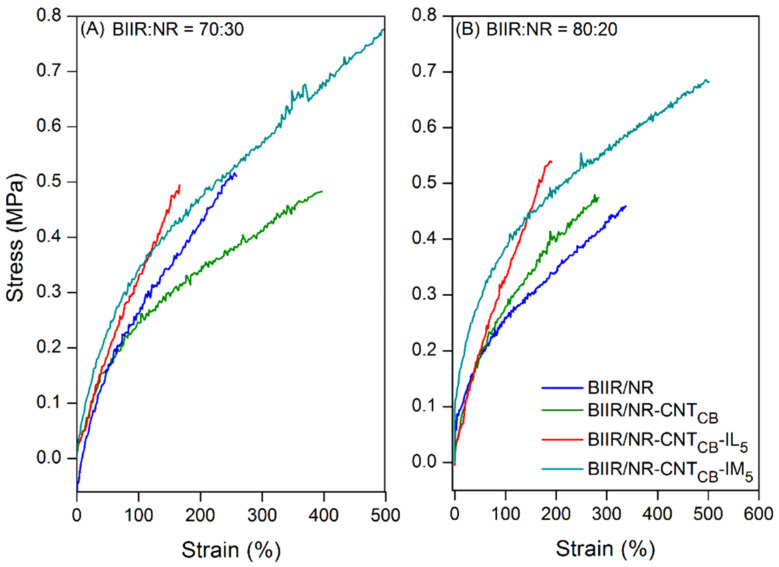
Stress–strain curves of pure rubbers and their composites with and without IL and IM modifiers, BIIR:NR ratios of 70:30 (**A**) and 80:20 (**B**) after healing propagation.

**Figure 10 polymers-13-00443-f010:**
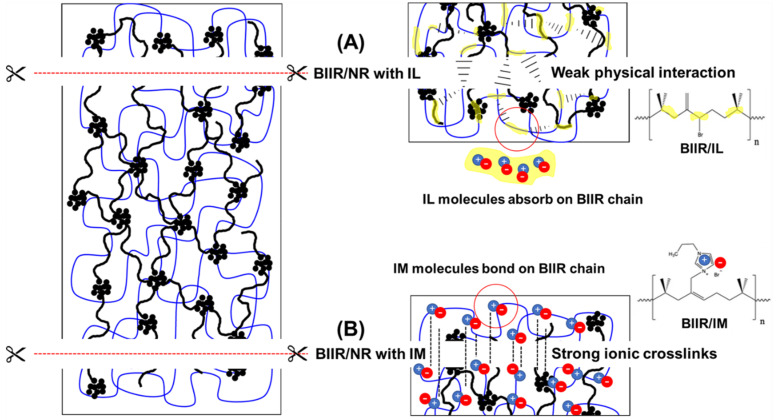
Proposed model of the BIIR/NR-CNT_CB_ composites filled with IL (**A**) and IM (**B**).

**Figure 11 polymers-13-00443-f011:**
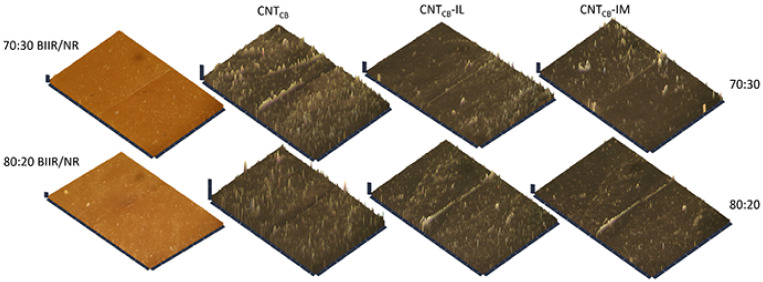
Morphologies of pure rubbers and their composites with and without IL and IM modifiers, BIIR:NR ratios of 70:30 and 80:20 after healing propagation.

**Figure 12 polymers-13-00443-f012:**
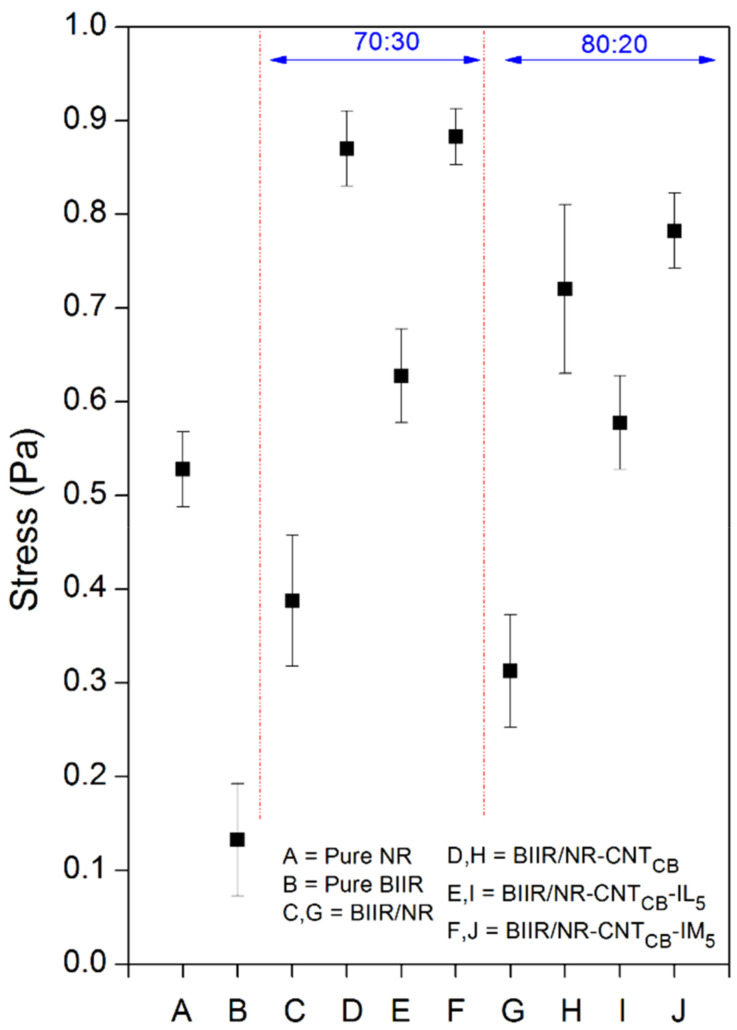
Puncture stress of pure rubbers and their composites with and without IL and IM modifiers, BIIR:NR ratios of 70:30 and 80:20.

**Figure 13 polymers-13-00443-f013:**
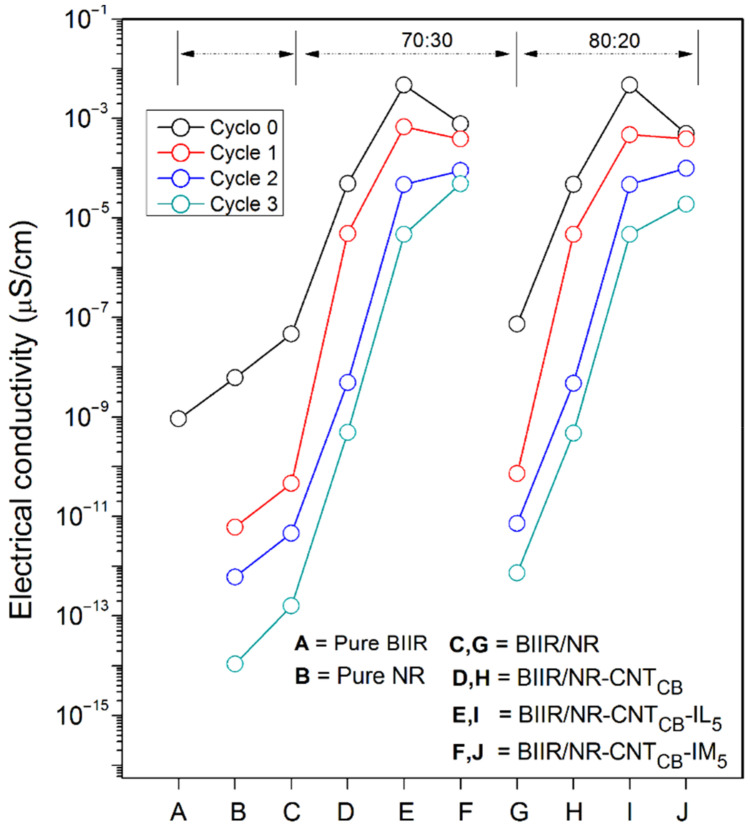
Electrical conductivity of pure rubbers and their composites with and without IL and IM modifiers, BIIR:NR ratios of 70:30 and 80:20 after recycling from 1, 2, and 3 rounds using the compression molding of 10,000 Pa.

**Figure 14 polymers-13-00443-f014:**
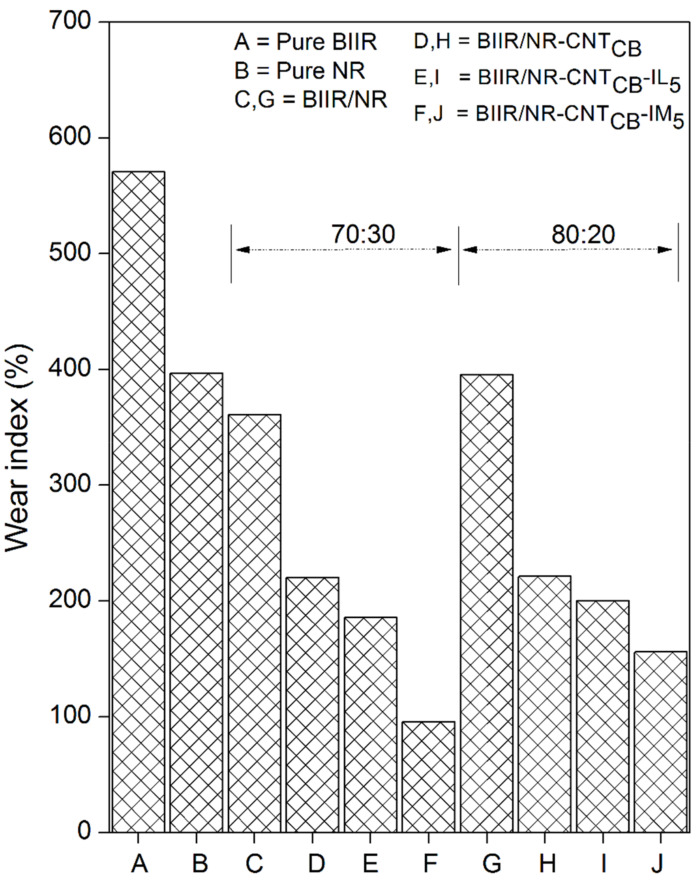
Wear index of pure rubbers and their composites with and without IL and IM modifiers, BIIR:NR ratios of 70:30 and 80:20.

**Figure 15 polymers-13-00443-f015:**
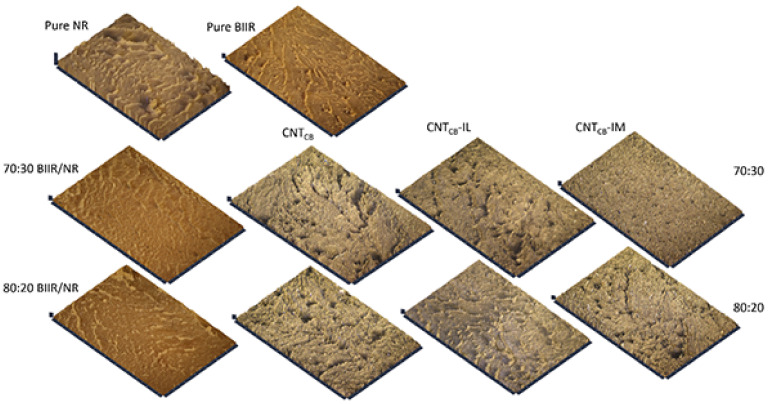
Surface roughness of pure rubbers and their composites with and without IL and IM modifiers, BIIR:NR ratios of 70:30 and 80:20 after Taber abrasion testing.

**Figure 16 polymers-13-00443-f016:**
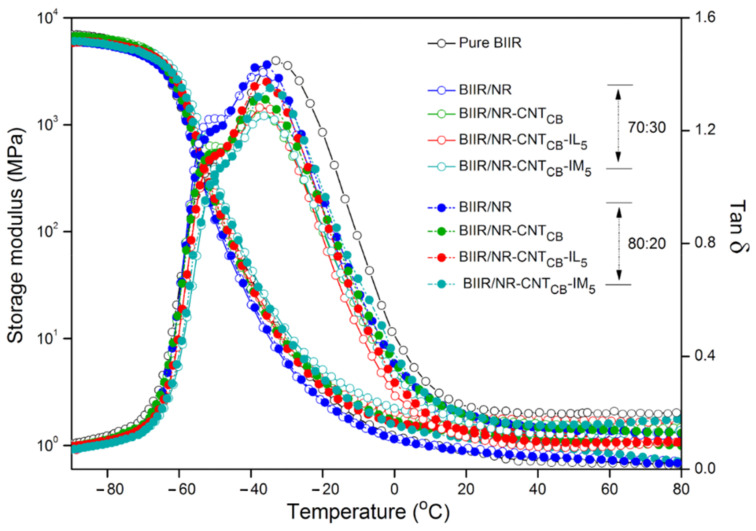
Storage modulus and Tan *δ* as a function of temperature of pure rubbers and their composites with and without IL and IM modifiers, BIIR:NR ratios of 70:30 and 80:20 after Taber abrasion testing.

**Table 1 polymers-13-00443-t001:** Chemicals formulation of pure rubbers and their composites.

Chemicals	Contents (phr*)	
**Modification of BIIR (COMPOUND I)**		
BIIR	100	**70 or 80**
IL or IM	5 and 10	
**Preparation of NR composites (COMPOUND II)**		
NR	100	**30 or 20**
CNT:CB	5:7.5	
ZnO	5	
Stearic acid	1	
MBTS	1	
Sulfur	2.5	

phr* = Part per hundred rubber.

**Table 2 polymers-13-00443-t002:** Cure characteristics of the pure rubbers and their composites.

Samples	*M_L_* (d.Nm)	*M_H_* (d.Nm)	*T*_90_ (min)	*T_S_*_1_ (min)	*M_H_–M_L_* (d.Nm)
Pure BIIR	1.14	1.40	−	−	0.27
Pure NR	1.07	5.39	3.28	1.28	4.32
**BIIR:NR ratio = 70:30**					
BIIR/NR	0.73	2.19	23.85	19.07	1.45
BIIR/NR-CNT_CB_	1.07	3.19	21.26	8.75	2.12
BIIR/NR-CNT_CB_-IL_5_	1.12	3.28	5.53	1.62	2.16
BIIR/NR-CNT_CB_-IM_5_	1.26	2.30	3.28	4.25	1.04
**BIIR:NR ratio = 80:20**					
BIIR/NR	0.86	2.09	23.55	21.58	1.24
BIIR/NR-CNT_CB_	1.07	2.68	21.74	16.40	1.62
BIIR/NR-CNT_CB_-IL_5_	1.15	2.61	10.25	4.04	1.45
BIIR/NR-CNT_CB_-IM_5_	1.32	1.72	2.52	−	0.39

**Table 3 polymers-13-00443-t003:** Mechanical properties in terms of 100%moduli, tensile strength and elongation at break of the pure rubbers and their composites before self-healing propagation.

Samples	100 % Modulus (MPa)	Tensile Strength (MPa)	Elongation at Break (%)
Pure NR	1.31 ± 0.02	14.50 ± 0.15	1412.45 ± 5.34
Pure BIIR	0.36 ± 0.05	0.12 ± 0.10	3124.45 ± 6.55
**BIIR:NR ratio = 70:30**			
BIIR/NR	0.35 ± 0.04	2.32 ± 0.12	1504.17 ± 14.56
BIIR/NR-CNT_CB_	0.48 ± 0.11	4.47 ± 0.25	1164.04 ± 23.16
BIIR/NR-CNT_CB_-IL_5_	0.96 ± 0.10	3.75 ± 0.10	551.82 ± 16.65
BIIR/NR-CNT_CB_-IM_5_	0.50 ± 0.06	3.31 ± 0.22	1743.16 ± 34.55
**BIIR:NR ratio = 80:20**			
BIIR/NR	0.35 ± 0.05	1.57 ± 0.15	1159.44 ± 21.76
BIIR/NR-CNT_CB_	0.44 ± 0.01	2.45 ± 0.51	1064.14 ± 62.98
BIIR/NR-CNT_CB_-IL_5_	0.70 ± 0.02	2.82 ± 0.32	564.75 ± 44.23
BIIR/NR-CNT_CB_-IM_5_	0.40 ± 0.04	1.79 ± 0.57	2476.55 ± 24.45

**Table 4 polymers-13-00443-t004:** Mechanical properties of the pure rubbers and their composites after self-healing propagation.

Samples	100 % Modulus (MPa)	Tensile Strength (MPa)	Elongation at Break (%)
Pure NR	−	−	−
Pure BIIR	0.33 ± 0.12	0.27 ± 0.15	674.65 ± 23.34
**BIIR:NR ratio = 70:30**			
BIIR/NR	0.34 ± 0.05	0.48 ± 0.12	396.59 ± 11.33
BIIR/NR-CNT_CB_	0.42 ± 0.06	0.51 ± 0.11	257.80 ± 32.23
BIIR/NR-CNT_CB_-IL_5_	0.50 ± 0.01	0.50 ± 0.08	166.10 ± 14.43
BIIR/NR-CNT_CB_-IM_5_	0.48 ± 0.14	0.78 ± 0.19	497.80 ± 15.18
**BIIR:NR ratio = 80:20**			
BIIR/NR	0.33 ± 0.04	0.46 ± 0.02	336.73 ± 32.33
BIIR/NR-CNT_CB_	0.41 ± 0.16	0.47 ± 0.05	283.77 ± 16.43
BIIR/NR-CNT_CB_-IL_5_	0.53 ± 0.07	0.54 ± 0.23	191.13 ± 18.55
BIIR/NR-CNT_CB_-IM_5_	0.50 ± 0.12	0.68 ± 0.45	502.25 ± 25.78

**Table 5 polymers-13-00443-t005:** Dynamic mechanical properties in terms of *T_g_*, *Tan δ_max_*, *Tan δ*_0_, and *Tan δ*_60_ of the pure rubbers and their composites.

Samples	*T_g_* (°C)	*Tan δ_max_*	*Tan δ* _0_	*Tan δ* _60_
Pure NR	−52.6	2.03	0.20	0.04
Pure BIIR	−33.0	1.45	0.49	0.20
**BIIR:NR ratio = 70:30**				
BIIR/NR	−36.1	1.41	0.35	0.13
BIIR/NR-CNT_CB_	−36.6	1.27	0.36	0.12
BIIR/NR-CNT_CB_-IL_5_	−36.6	1.28	0.26	0.10
BIIR/NR-CNT_CB_-IM_5_	−36.4	1.25	0.37	0.11
**BIIR:NR ratio = 80:20**				
BIIR/NR	−35.6	1.43	0.36	0.14
BIIR/NR-CNT_CB_	−36.0	1.33	0.41	0.13
BIIR/NR-CNT_CB_-IL_5_	−36.1	1.36	0.31	0.10
BIIR/NR-CNT_CB_-IM_5_	−5.7	1.32	0.43	0.12

## Data Availability

The data presented in this study are available on request from the corresponding author.

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
