# Peer review of "Combination of Self-Healing Butyl Rubber and Natural Rubber Composites for Improving the Stability"

_polymers, 2021, doi:10.3390/polym13030443_

Round 1
Reviewer 1 Report
The manuscript by Chumnum and co-workers deals with the investigation of self-healing composites prepared from the combination of bromobutyl rubber (BIIR) and natural rubber (NR) blends filled with carbon-based fillers. Moreover, the authors modified BIIR with ionic liquid (IL) and butylimidazole (IM) before blending to reach optimal self-healing propagation. In fact, the application and study of ionic liquids in terms of the filler dispersion, curing efficiency and other activities in the rubber systems is now under high consideration. Therefore, the topic taken by the authors is of great importance for the extending the knowledge about the blending and self-healing process in rubber technology. Furthermore, many important parameters characteristic for tire products have been discussed in the paper which is important from the technological point of view. The manuscript has a clear structure that make it easy to understand it, and the methods were selected correctly to discuss the research hypothesis of the article. In my opinion, the abovementioned work can be considered for publication in Polymers journal after improving some issues:
1) Generally, scientific works dealing with hybrid fillers consists of compounds with different filler ratios. Why did the authors decide to use CNT: CB = 5:7.5 phr ratio? Have you made any optimization of such system before? (page 3, line 111)
2) Did you control the mass temperature in the Brabender when adding the curatives into the elastomer compounds? The friction might lead to the increase of temperature and caused scorch process. (page 3, line 115)
3) How many specimens of each composites have been used for tensile properties and abrasion resistance measurements?
4) The authors decided to use the specific surface area of the fillers to determine the average thickness of rubber layer bound to the filler surfaces (page 4, line 167). How did you know the SSA values for these fillers (from the manufacturer or based on the low-temperature nitrogen adsorption measurements?)? Can you add SSA values for both fillers into manuscript? Perhaps, it is worth considering this parameter when discussing the reinforcing effect of the fillers.
5) I suggest using term “macromolecules” instead of “molecules” when dealing with polymeric materials (page 6, line 239).
5) The quality of the figures must be improved as many of them are poorly legible.
7) The authors explained the reduced TS1 and T90 parameters upon presence of IL in the composition by improved filler dispersion and superior thermal conductivity of the carbon fillers (page 7, line 278). The reviewer agrees with that statement; however, the authors can also consider catalytic effect of the ionic liquids on interface reactions between elastomer and carbon filler during crosslinking process (e.g., Polymer Testing, 2020, 81, 106222).
8) The authors also stated that “…existing IL (i.e. liq-314 uid phases) in the matrix makes more defects during extension and causes the material easily breakable, lowers the tensile strength and elongation”. Has such a phenomenon been already described in the literature? (page 9, line 314).
9) The results presented in the Figure 5 are not discussed in the manuscript.
10) In Table 3 and 4, the measurement errors must be added.
11) Parenthesis symbols in tables should be corrected.
Author Response
Comments and Suggestions for Authors: Reviewer 1
The manuscript by Chumnum and co-workers deals with the investigation of self-healing composites prepared from the combination of bromobutyl rubber (BIIR) and natural rubber (NR) blends filled with carbon-based fillers. Moreover, the authors modified BIIR with ionic liquid (IL) and butylimidazole (IM) before blending to reach optimal self-healing propagation. In fact, the application and study of ionic liquids in terms of the filler dispersion, curing efficiency and other activities in the rubber systems is now under high consideration. Therefore, the topic taken by the authors is of great importance for the extending the knowledge about the blending and self-healing process in rubber technology. Furthermore, many important parameters characteristic for tire products have been discussed in the paper which is important from the technological point of view. The manuscript has a clear structure that make it easy to understand it, and the methods were selected correctly to discuss the research hypothesis of the article. In my opinion, the abovementioned work can be considered for publication in Polymers journal after improving some issues:
1) Generally, scientific works dealing with hybrid fillers consists of compounds with different filler ratios. Why did the authors decide to use CNT: CB = 5:7.5 phr ratio? Have you made any optimization of such system before? (page 3, line 111)
ANS The reviewer is right. This ratio of CNT:CB at 5:7.5 was proven previously by our researches teams. This ratio indicated the optimum value of tensile properties, dynamic mechanical properties and electrical conductivity. However, for avoid the misunderstanding, we added the reference No.[23] in the “Preparation of self-healing BIIR/NR composites” section of the manuscript as seen in “BLUE” letters.
2) Did you control the mass temperature in the Brabender when adding the curatives into the elastomer compounds? The friction might lead to the increase of temperature and caused scorch process. (page 3, line 115)
ANS Yes, we did a carefully control of the mixing temperature. Also, we used here the mixing torque and ML value to make sure that there has no pre-crosslinking among rubber molecules since it has no strong increased of the mixing torque after adding the curatives while all the ML values have no significantly changed.
3) How many specimens of each composites have been used for tensile properties and abrasion resistance measurements?
ANS Thank you the reviewer for the effective question. We used 5 samples for tensile measurement and 3 repeating samples from abrasion testing. Therefore, we added the specific errors in the Tables 3 and 4 of the tensile properties, while we displayed abrasion resistance of the composites relating the showing abrasion morphologies. The revised version are indicated in “BLUE” letters of the manuscript for Tables 3 and 4, and for “Characterization” section.
4) The authors decided to use the specific surface area of the fillers to determine the average thickness of rubber layer bound to the filler surfaces (page 4, line 167). How did you know the SSA values for these fillers (from the manufacturer or based on the low-temperature nitrogen adsorption measurements?)? Can you add SSA values for both fillers into manuscript? Perhaps, it is worth considering this parameter when discussing the reinforcing effect of the fillers.
ANS We received the values from the specification of the manufacturer based on the BET processes. However, we added both values into the manuscript in “Characterization” section as indicated in “BLUE” letters.
5) I suggest using term “macromolecules” instead of “molecules” when dealing with polymeric materials (page 6, line 239).
ANS Thank you very much the reviewer, we do appreciate for your suggestion. Thus, we revised all the typos appearing in manuscript as seen in “BLUE” letters.
6) The quality of the figures must be improved as many of them are poorly legible.
ANS We have revised Figures 2 and 10 to be higher resolution and easier for following.
7) The authors explained the reduced TS1 and T90 parameters upon presence of IL in the composition by improved filler dispersion and superior thermal conductivity of the carbon fillers (page 7, line 278). The reviewer agrees with that statement; however, the authors can also consider catalytic effect of the ionic liquids on interface reactions between elastomer and carbon filler during crosslinking process (e.g., Polymer Testing, 2020, 81, 106222).
ANS Thank you very much the reviewer for the effective advices. We have carefully read the suggested manuscript and do agree with the explanation of the authors. Therefore, we revised the discussion in “Cure characteristics” section and added also the suggested reference into “Reference” section as seen in “BLUE” letters of the manuscript.
8) The authors also stated that “…existing IL (i.e. liq-314 uid phases) in the matrix makes more defects during extension and causes the material easily breakable, lowers the tensile strength and elongation”. Has such a phenomenon been already described in the literature? (page 9, line 314).
ANS The reviewer is right. The effect of IL on lowering mechanical properties has proven previously from our research teams. It was found that the IL existing in rubber matrix are acted as the liquid phase inside rubber matrix and caused easily breakage of the rubber molecules during extension. Therefore, we added the references No. [45], [46] in the “Relations of mechanical, dynamic mechanical and morphologies” section as indicated in “BLUE” letters of the manuscript.
9) The results presented in the Figure 5 are not discussed in the manuscript.
ANS Thank you the reviewer for the requirement. We added the discussion of Figure 5 following the suggestion in “Relations of mechanical, dynamic mechanical and morphologies” section as also seen below:
“Considering in Figure 5, it is clearly seen that the storage modulus of the BIIR, NR and BIIR/NR increases with the addition of CNT-CB hybrid filler. This is due to the reinforcement efficiency of the filler to rubber matrix. In addition, it is seen that the highest storage modulus had found from the composites with IM, while the one with IL showed poor properties. This affirms the efficiency of the chemical crosslinking among BIIR macromolecules after mixing with NR which is also the rationale of slight higher storage modulus of the composites using BIIR:NR ratio of 80:20 than 70:30.”
The revised statements are displayed in “BLUE” letters of the manuscript.
10) In Table 3 and 4, the measurement errors must be added.
ANS Thank you very much the reviewer. We do agree with the suggestion and therefore we have revised Tables 3 and 4 by adding the measurement errors of each sample.
11) Parenthesis symbols in tables should be corrected.
ANS Parenthesis symbols of the header of each Table have already revised following the suggestion.

Reviewer 2 Report
The manuscript related to the composite of self-healing BR and NR has been reviewed. I appreciate the authors for studying this worthy topic and performing detailed analysis to justify their hypothesis. Though the article is well written, a few changes are extremely important to be considered for publication.
- Please check the centigrade symbol, i.e., page 4, line 148.
- The quality of Figure 2 is unacceptable. Please provide a clear image.
- Also, in Figure 2, make sure to mark each section.
- Please mark the numbers for all subsections.
- I appreciate authors performing detailed FTIR analysis. However, while mentioning any FTIR peak, it is highly recommended to cite the relevant source, unless the FTIR peak is very well known in the given field.
- Please check the table header of Table 2.
- Please provide the statical error within the data in the tables.
- Once again, please check the table header of Table 3. Check the direction of parenthesis
- Figure 10 is a bit confusing; please mark A and B. It is not easy to understand, and the two mechanisms are not presented in a good manner. Therefore, it is advised to either delete this section and Figure or draw a better one.
- Reference 50 seems not correct; considering adding article number, i.e., 1600385. The same applies to Reference 51.
Author Response
Comments and Suggestions for Authors: Reviewer 2
The manuscript related to the composite of self-healing BR and NR has been reviewed. I appreciate the authors for studying this worthy topic and performing detailed analysis to justify their hypothesis. Though the article is well written, a few changes are extremely important to be considered for publication.
1. Please check the centigrade symbol, i.e., page 4, line 148.
ANS Thank you very much the reviewer, we have revised the presented symbol as seen in the “BLUE” letters of the manuscript.
2. The quality of Figure 2 is unacceptable. Please provide a clear image.
ANS Resolution of Figure 2 has revised.
3. Also, in Figure 2, make sure to mark each section.
ANS Thank you very much the reviewer, we have revised Figure 2 as the suggestion.
4. Please mark the numbers for all subsections.
ANS The numbers of all sections are added in the manuscript as displayed in “BLUE” letters.
5. I appreciate authors performing detailed FTIR analysis. However, while mentioning any FTIR peak, it is highly recommended to cite the relevant source, unless the FTIR peak is very well known in the given field.
ANS Indication of the assignment from the FTIR spectra has revised with addition of references as seen in the “BLUE” letters of “Interactions of IL and IM with rubber molecules” section.
6. Please check the table header of Table 2.
ANS The headers of T90 and Ts1 units have edited following the suggestion.
7. Please provide the statical error within the data in the tables.
ANS We added the standard errors of the tensile properties as following in Tables 3 and 4, relatively.
8. Once again, please check the table header of Table 3. Check the direction of parenthesis
ANS The headers in Table 3 have already revised as can be seen in the manuscript.
9. Figure 10 is a bit confusing; please mark A and B. It is not easy to understand, and the two mechanisms are not presented in a good manner. Therefore, it is advised to either delete this section and Figure or draw a better one.
ANS Thank you very much the reviewer for the effective requirement. We have revised Figure 10 for more understandable as seen in the manuscript.
10. Reference 50 seems not correct; considering adding article number, i.e., 1600385. The same applies to Reference 51.
ANS Thank you the reviewer for the suggestion. We have revised references No. [50] and [51] as indicated in “BLUE” letters of the manuscript (i.e. in “References” section).
According to the revised version of the manuscript the numbers of the revised references have been changed to be [51] and [52], respectively.

Round 2
Reviewer 1 Report
The authors responded in detail to all my objections. In my opinion, the manuscript in its current form can be accepted for publication.
Reviewer 2 Report
The authors have improved the manuscript as per the reviewer's guidance; therefore, the manuscript might be considered for publication in its present form.